# Effect of Preoperative Single-Inhaler Triple Therapy on Pulmonary Function in Lung Cancer Patients with Chronic Obstructive Pulmonary Disease and FEV_1_ < 1.5 L

**DOI:** 10.3390/cancers17111803

**Published:** 2025-05-28

**Authors:** Takahiro Homma, Hisashi Saji, Yoshifumi Shimada, Keitaro Tanabe, Koji Kojima, Hideki Marushima, Tomoyuki Miyazawa, Hiroyuki Kimura, Hiroki Sakai, Kanji Otsubo, Takayuki Hatakeyama, Norifumi Kakizaki, Tomoshi Tsuchiya, Kei Morikawa, Masamichi Mineshita

**Affiliations:** 1Department of Thoracic Surgery, St. Marianna University School of Medicine, Kanagawa 216-8511, Japan; 2Division of Thoracic Surgery, Kurobe City Hospital, Toyama 938-8502, Japan; 3Division of Thoracic Surgery, Joetsu General Hospital, Niigata 943-0172, Japan; 4Department of Thoracic Surgery, University of Toyama, Toyama 930-0194, Japan; 5Division of Respiratory Medicine, St. Marianna University School of Medicine, Kanagawa 216-8511, Japan

**Keywords:** chronic obstructive pulmonary disease, asthma, video-assisted thoracoscopic surgery, lung resection, lung cancer, poor pulmonary function

## Abstract

This study investigated the impact of preoperative single-inhaler triple therapy on surgical treatment selection in lung cancer patients with COPD and FEV_1_ < 1.5 L. All patients had baseline FEV_1_ < 1.5 L. Triple therapy was initiated 2 weeks preoperatively and continued until 3 months postoperatively. Triple-inhaler therapy improved FEV_1_ to ≥1.5L in 63.4% of patients, significantly increasing radical resection rates (88.5% vs. 40.0%, *p* = 0.0030) compared with patients whose FEV_1_ remained <1.5 L post-treatment. Postoperative pneumonia and home oxygen therapy were observed only in patients whose FEV_1_ remained <1.5 L post-treatment. Preoperative triple therapy improved respiratory function, increasing the feasibility of radical resection with a low incidence of postoperative complications.

## 1. Introduction

In the standard surgical procedure for lung cancer, which is lobectomy, a preoperative forced expiratory volume in one second (FEV_1_) < 1.5 L is recognized as a risk factor for perioperative complications [1]. Considering the risk of standard surgery when FEV_1_ < 1.5 L, a decision-making process on whether to proceed with the surgery, opt for palliative surgery, or choose alternative non-surgical treatments arises. Generally, surgical intervention tends to offer better treatment outcomes if resection is feasible [2]. However, due to the risks of postoperative complications and home oxygen therapy [3], patients with low respiratory function may opt for limited resection or non-surgical treatments. Additionally, even in early-stage non-small cell lung cancer patients with chronic obstructive pulmonary disease (COPD), the risks of poor overall survival and recurrence-free survival are significantly increased [4].

Previous reports have shown that preoperative interventions for untreated COPD help maintain good respiratory function [5,6,7,8]. We previously reported the effectiveness of preoperative inhalation therapy for untreated COPD patients with concurrent lung cancer [9], and showed that initiating inhalation therapy 2 weeks before surgery resulted in postoperative respiratory function equivalent to pre-lung resection levels (FEV_1_: −0.05 vs. −0.25 mL, *p* = 0.0026) and better respiratory function than predicted by the formula (+0.33 vs. +0.04 mL, *p* < 0.0001). Additionally, the inhalation therapy using the combination of umeclidinium and vilanterol (UMEC/VI) was found to be favorable compared to tiotropium (TIO) monotherapy (160 vs. 7 mL, *p* = 0.0005). Considering these findings, introducing preoperative inhalation therapy proactively, particularly UMEC/VI, is deemed effective for untreated COPD patients with concurrent lung cancer. The fact that 31.8% of lung cancer patients have concurrent, untreated COPD suggests the potential impact of appropriate therapeutic interventions for COPD on treatment choices for lung cancer. This highlights the importance of identifying and addressing untreated COPD in patients diagnosed with lung cancer.

In recent years, single-inhaler triple therapy, involving long-acting muscarinic antagonists, long-acting beta-agonists, and inhaled corticosteroids (ICSs), has been reported to be effective in the treatment of COPD and asthma [10,11,12,13]. However, evidence on the use of single-inhaler triple therapy during the perioperative period is currently lacking.

Thus, we hypothesized that introducing single-inhaler triple therapy for untreated COPD patients with FEV_1_ < 1.5 L could influence the choices in lung cancer treatment. This study aimed to investigate the effects, safety, and impact on treatment choices of fluticasone furoate (FF)/UMEC/VI in untreated COPD patients with FEV_1_ < 1.5 L and concurrent lung cancer. The novelties of this study are (1) introducing single-inhaler triple therapy during the perioperative period of lung resection, (2) use of FF/UMEC/VI, and (3) examining postoperative pneumonia associated with ICSs.

## 2. Materials and Methods

### 2.1. Ethical Statement

The study was conducted in accordance with the Declaration of Helsinki (as revised in 2013). The study was centrally approved by the Ethics Committee of Joetsu General Hospital (Niigata, Japan; J-183). The requirement for informed consent was waived due to the retrospective design of the study.

### 2.2. Study Design and Patients

Lung cancer patients with COPD admitted to the Joetsu General Hospital, Toyama University Hospital (Toyama, Japan), and Kurobe City Hospital (Toyama, Japan) for elective lung surgery between April 2016 and May 2023 were included.

In this study, COPD was defined based on three criteria: (1) a history of exposure to risk factors including secondhand smoking or the inhalation of particulate matter in the living environment, regardless of symptoms [14,15,16]; (2) FEV_1_/forced vital capacity ratio <0.70 on spirometry after bronchodilator administration; and (3) exclusion of other conditions that could cause airflow obstruction [17]. This study included patients diagnosed with asthma–COPD overlap (ACO), as defined by the combined guidelines of the Global Initiative for Asthma (GINA) and Global Initiative for Chronic Obstructive Pulmonary Disease (GOLD) [18]. ACO patients were identified by having COPD along with features of asthma, such as: (1) more frequent exacerbations responsive to treatment; (2) rapid progression and high treatment needs; (3) history of doctor-diagnosed asthma, allergies, or family history of asthma; (4) respiratory symptoms like persistent exertional dyspnea with potential variability; (5) symptom onset in childhood or early adulthood. The exclusion criteria were patients with FEV_1_ ≥ 1.5 L before COPD treatment, treated COPD, asthma alone, pneumonectomy, and emergency surgery; with or suspected of bronchial obstruction by tumor; aged ≤19 years; and with missing data. Patients with ACO were not excluded from the present study.

### 2.3. COPD Treatments

The COPD treatment policy at our medical centers was as follows. A therapeutic drug was prospectively prescribed. Patients with FEV_1_ < 1.5 L were initiated on triple therapy, including FF 100 μg, UMEC 62.5 μg, and VI 25 μg, administered using the same once-daily inhalation device, Ellipta. Starting in March 2021, single-inhaler triple therapy with FF/UMEC/VI was prescribed. Before that, triple therapy was administered with a combination of FF and UMEC/VI or FF/VI and UMEC using the same Ellipta device. Each of these drugs was administered from 2 weeks before surgery and continued until 3 months after surgery at least. An expectorant was used in all cases.

### 2.4. Preoperative Tests

Simple chest radiography, contrast-enhanced computed tomography, electrocardiograms, and blood, urine, and pulmonary function tests were performed in all cases. Spirometry was performed after bronchodilator administration. In cases with lung cancer, cerebral magnetic resonance imaging and positron emission tomography were also performed. Patients aged >80 years with one or more risk factors for coronary artery disease underwent preoperative cardiac echography and cardiac stress testing.

### 2.5. Surgical Strategy

Radical lobectomy and node dissections were performed in patients with FEV_1_ ≥ 1.5 L after COPD treatment. For well-differentiated lung adenocarcinomas ≤ 2 cm with a tumor stage of cT1aN0M0 or less, radical segmentectomy was performed, even if FEV_1_ was ≥1.5 L [19]. If FEV_1_ was <1.5 L after COPD treatment, patients were informed about the incidence of perioperative complications based on the extent of lung resection and the risk of local recurrence compared with that of palliative surgery. Radical lobectomy was performed if the patient actively selected this procedure or if the surgeon deemed it necessary. Palliative surgery was performed if the patient was reluctant to undergo lobectomy or if the risk of perioperative complications was very high. Radiation therapy was chosen when surgery was deemed high-risk due to strong patient preferences or comorbidities, regardless of respiratory function. The final treatment decision was made by the patients after a thorough explanation by the physician. In addition, preoperative pulmonary training using TRI-BALL^TM^ (Medtronic, Dublin, Ireland) as an exercise device was performed in all surgical patients. No postoperative rehabilitation was performed. Radiation therapy was administered at the patient’s request or if the surgeon deemed radiation therapy more appropriate than surgery. All patients who smoked were advised to stop smoking and provided treatment for smoking cessation, regardless of the administered COPD treatment.

### 2.6. Surgical Procedure

General anesthesia was maintained using single-lung ventilation with a double-lumen endotracheal tube. Patients were placed in the lateral decubitus position. The surgical approach involved video-assisted thoracoscopic surgery (VATS). Patients underwent four-port VATS with three 5-mm ports and one 10-mm port for lobectomy and segmentectomy from April 2016 and uniportal VATS from July 2018 [20]. A thoracoscope was used with a 30° 5- or 10-mm camera. During specimen extraction, a one-port incision was extended to approximately 3 cm in multiportal VATS patients.

### 2.7. Variables and Assessments

Patient and surgical characteristics, as well as follow-up parameters, were recorded from the preoperative period until 3 months after surgery, including age, sex, past medical history, smoking history, body mass index (BMI), respiratory function as measured by spirometry, lung cancer and COPD diagnosis, disease side, tumor size, interstitial pneumonia, procedure type (partial resection, segmentectomy, lobectomy), location (upper including right middle and lower lobe), number of resected lung segments, surgical approach (uniportal VATS, multiportal conventional VATS, thoracotomy), intraoperative bleeding, operative time, chest tube duration, complications (e.g., prolonged air leak defined as an air leak lasting for >5 days), as well as inhalation therapy and its side effects. Complications were defined as any deviation from the normal postoperative course and graded according to the Clavien–Dindo classification. For the definition of radical resection, we classified lobectomy and segmentectomy as radical for cT1bN0M0 Stage IA2 or less. For lung cancer of higher stages, lobectomy was defined as radical resection. All partial resections, regardless of stage, were categorized as palliative surgery.

Pulmonary function tests were performed before administering the prescription, 2 weeks after the prescription, and 3 months after surgery. The following formula was used to derive postoperative predicted respiratory function: predicted postoperative FEV_1_ = preoperative FEV_1_ × (19 − segments to be removed/19) [21]. Side effects of inhalation therapy were measured in cases of thirst, urination difficulty, eye symptoms, palpitation, arrhythmia, headache, nausea, vomiting, and low blood pressure, when no causes other than inhalation therapy were observed.

### 2.8. Data Management and Statistical Analysis

Data were retrospectively collected at Joetsu General Hospital, Toyama University Hospital, and Kurobe City Hospital, which partially overlapped with those used in our previous study [9]. However, it should be noted that the current study focused on FF/UMEC/VI and COPD patients with FEV_1_ < 1.5 L, whereas the previous study centered on UMEC/VI and all untreated COPD patients.

For the univariate analysis, intergroup differences were evaluated using the non-parametric Wilcoxon rank-sum test. The χ^2^ or Fisher’s exact test was used to compare categorical variables as appropriate. Statistical significance was defined as a two-sided *p*-value of <0.05. Continuous variables are presented as mean ± standard deviation for normally distributed data and median with interquartile range for non-normally distributed data. Categorical variables are presented as sample size and percentage (*n* (%)). All statistical analyses were performed using JMP version 15.0 (SAS Institute Inc., Cary, NC, USA).

## 3. Results

Among 675 lung cancer patients, 214 (31.7%) had COPD/ACO, of whom 35 (16.4%) that had undergone treatment interventions were excluded from this study. Of the remaining 179 patients, 41 had an FEV_1_ < 1.5 L (Figure 1, Table 1).

### 3.1. Analysis of All Patients Undergoing Lung Resection

Introducing single-inhaler triple therapy resulted in a significant improvement in FEV_1_, with a median increase of 210 mL [0.11–0.38]. In particular, 6 out of 41 patients (14.6%) showed an improvement of more than 400mL in FEV_1_ after inhalation. Among the 41 cases, 26 (63.4%) achieved an FEV_1_ ≥ 1.5 L after inhalation therapy, while 15 (36.6%) remained at <1.5 L (Figure 2). There were significant differences in pre-intervention FEV_1_ (1.37 vs. 1.14L, *p* = 0.0050) and treatment-induced increase in FEV_1_ (0.29 vs. 0.14 L, *p* = 0.028) between patients achieving FEV_1_ ≥ 1.5 L and those with FEV_1_ remaining at <1.5 L after COPD intervention. Patients with FEV_1_ ≥ 1.5 L had a significantly lower Brinkman index than those with FEV_1_ < 1.5 L (840 vs. 1120, *p* = 0.0058). The radical resection rate was significantly higher in patients with FEV_1_ ≥ 1.5 L (88.5% vs. 40.0%, *p* = 0.0030). There were no significant differences in other perioperative factors between the two groups. Postoperative pneumonia and initiation of home oxygen therapy (HOT) occurred in two (4.9%) and one (2.4%) case, respectively, all of which were patients with FEV_1_ < 1.5 L. There were no patients undergoing radiation therapy or postoperative deaths.

### 3.2. Analysis of Patients Undergoing Anatomical Lung Resection

In the subgroup analysis of 33 patients undergoing anatomical lung resection, 23 (69.7%) and 10 (30.3%) had FEV_1_ ≥ 1.5 L and FEV_1_ < 1.5 L, respectively (Table 2). Among patients with FEV_1_ ≥ 1.5 L, similar to the overall comparison, significant differences were observed in the Brinkman index (840 vs. 1100, *p* = 0.013), pre-inhalation FEV_1_ (1.37 vs. 1.13, *p* = 0.0013), and the radical resection rate (100.0% vs. 60.0%, *p* = 0.0051). Additionally, the postoperative complications in patients with FEV_1_ ≥ 1.5 L were significantly lower (13.0% vs. 50.0%, *p* = 0.036).

In patients undergoing anatomical lung resection, there was a significant increase in FEV_1_ post-inhalation therapy (1.55 vs. 1.26 L, *p* < 0.0001) (Figure 3). The predicted postoperative FEV_1_ before intervention was significantly lower than that after intervention (1.06 vs. 1.31 L, *p* < 0.0001) and the actual FEV_1_ at 3 months postoperatively (1.06 vs. 1.31 L, *p* < 0.0001) (Figure 4). The FEV_1_ at 3 months postoperatively was significantly better than the pre-inhalation therapy level (1.31 vs. 1.26 L, *p* = 0.042) (Figure 3).

## 4. Discussion

In this study, preoperative single-inhaler triple therapy in 41 lung cancer patients with untreated COPD/ACO and FEV_1_ < 1.5 L resulted in a significant improvement in FEV_1_, with an increase of 210 mL [0.11–0.38] compared to pre-intervention levels, and 26 of them (63.4%) achieved an FEV_1_ ≥ 1.5 L after inhalation. Patients with a higher Brinkman index tended to have an FEV_1_ < 1.5 L (840 vs. 1120, *p* = 0.0058). The radical resection rate was significantly higher in patients who achieved an FEV_1_ ≥ 1.5 L after inhalation (88.5% vs. 40.0%, *p* = 0.0030). Postoperative pneumonia occurred within 3 months in 2 out of 41 cases (4.9%), both of which were patients with FEV_1_ < 1.5 L. Moreover, in patients undergoing anatomical lung resection, single-inhaler triple therapy not only significantly improved respiratory function (1.26 vs. 1.55L, *p* < 0.0001), but also resulted in a significantly better FEV_1_ at 3 months postoperatively compared to the predicted pre-intervention FEV_1_ (1.31 vs. 1.06L, *p* < 0.0001). The FEV_1_ at 3 months postoperatively was significantly better than pre-inhalation levels (1.31 vs. 1.26L, *p* = 0.042).

The findings of this study suggest that preoperative single-inhaler triple therapy for lung cancer patients with untreated COPD/ACO and FEV_1_ < 1.5 L may enhance the feasibility of radical resection. Additionally, ICSs are not considered to always increase postoperative pneumonia. Even after anatomical lung resection, maintaining respiratory function at or above the pre-inhalation therapy level at 3 months postoperatively is considered possible. In cases with a higher Brinkman index (>1100), FEV_1_ might not significantly increase in many instances despite triple therapy.

In COPD, respiratory function worsens gradually, symptoms appear gradually, and the diagnosis requires time. The relationship between perioperative outcomes and COPD is considered poor not only for lung cancer, but also for diseases of other organs [22,23,24,25,26,27,28].

In our previous study, preoperative inhalation therapy (UMEC/VI or TIO) for untreated COPD patients with lung cancer enhanced respiratory function preoperatively (median 120 mL) [9]. Direct comparison with this study was not feasible due to differences in patient backgrounds. However, this study demonstrated similar effectiveness of preoperative inhalation therapy in patients with low FEV_1_. Furthermore, the improvement in respiratory function observed with triple therapy was substantial (median 210 mL), which is consistent with existing reports [10,11,12,13]. This improvement may be attributable to the additional agent in the triple therapy. Future studies with a prospective comparative design, such as a randomized controlled trial, are needed to confirm these findings and investigate the underlying mechanisms.

Pulmonary resection in COPD patients is associated with postoperative respiratory complications (atelectasis, persistent air leak, pneumonia) [26,27,28]. Single-inhaler triple therapy increases the likelihood of choosing radical surgery; however, concerns remain that ICS might increase postoperative pneumonia risk, given that ICS discontinuation is associated with reducing the risk of severe pneumonia [29,30]. It is well-established that ICS can increase the risk of pneumonia. For instance, the adjusted rate ratio of hospitalization for pneumonia associated with the current use of inhaled corticosteroids was 1.70 (95% confidence interval [CI], 1.63–1.77) [31]. Furthermore, the risk of pneumonia may be influenced by the dosage of inhaled corticosteroids; the rate ratio of hospitalization for pneumonia was greatest with the highest doses of inhaled corticosteroids, equivalent to fluticasone at 1000 µg/day or more (rate ratio, 2.25; 95% CI, 2.07–2.44) [31]. In our study, a low dose of ICS was used, and it is plausible that using higher doses could alter the side effect profile. Additionally, regarding postoperative pneumonia, a direct causal relationship between ICS use and perioperative complications cannot be definitively concluded from our data. However, the overall incidence of pneumonia complications observed in our cohort was low. While the diagnosis of COPD was primarily evaluated by spirometry after bronchodilator administration, it is important to note that some patients were nonsmokers, making it difficult to definitively rule out asthma [32]. In this study, only two patients (4.9%) who had FEV_1_ < 1.5 L after inhalation developed pneumonia. Although a direct comparison between UMEC/VI and FF/UMEC/VI was not conducted in this study, the incidence of complications did not differ significantly compared to the previous study on UMEC or VI (32.8%) [9]. The impact of triple therapy on postoperative complications was low, which is consistent with previous reports that the risk of postoperative complications increases as FEV_1_ decreases [1,26,27,28]. Segmentectomy for early-stage small lung cancer has gained increasing recognition in recent years, leading to its widespread adoption [19,33]. This may contribute to reducing the risk of postoperative complications.

Dysphonia is a recognized adverse effect associated with inhalation therapy [30]. No patients in this study experienced this complication during the observation period. However, it is important to ensure that patients are informed about this potential adverse effect and receive comprehensive guidance on preventive strategies [34]. Postoperative dysphonia may also occur following upper mediastinal lymph node dissection [35]. Consequently, differentiating between the influence of inhalation therapy and the surgical intervention is essential.

ACO diagnosis is not easy [18], and approximately 20% of COPD patients have ACO [36]. Since FEV_1_ improved more than 400 mL after inhalation in 6 out of 41 patients (14.6%), the incidence of ACO was considered similar to that in previous reports. In a previous study using TIO or UMEC/VI, the incidence of ACO was 9.8% (6 in 61 patients) [9]. Therefore, triple therapy with FF/UMEC/VI may be more effective, suggesting the potential to identify latent ACO patients. The results of this study can be considered one of the indications for the significance of preoperative single-inhaler triple therapy in patients with low FEV_1_ lung cancer.

There are negative opinions in the medical community regarding the costs of interventions for asymptomatic COPD patients. It is also known that postoperative respiratory function often plateaus around three months. Therefore, our study intentionally limited the duration of inhaler use to three months, and we did not intend for longer-term administration of the medication within the scope of this research. Although the costs were not considered in this study, given the high treatment costs for recurrent lung cancer [37], the value of providing opportunities for radical surgery without increasing perioperative complications would be high. Therefore, the significance of introducing preoperative single-inhaler triple therapy in untreated COPD patients was deemed considerable.

We assumed that asymptomatic COPD patients may forget to use their drugs. Therefore, we selected drugs that only had to be inhaled once a day. Triple therapy that can be administered once daily is likely to become increasingly popular [38]. In this study, since the perioperative period was relatively short, inhalation therapy was continued in all cases, and no significant side effects were reported. The patient population included many older men, but none of the patients complained of worsening dysuria. Some patients reported choking due to inhalation and associated exacerbation of transient pain, for which they did not wish to continue therapy for more than 3 months postoperatively. During the study period, only FF/UMEC/VI could be prescribed as single-inhaler triple therapy, but currently, there are other options [11,12]. Differences in inhaled medications and devices may lead to individual variations, and in some cases, changing the inhaled medication may be beneficial [39]. Currently, other triple therapies have emerged, which may become additional options [11,12,40].

### Limitations

The limitations of this study include its retrospective design, sample size, selection of inhaled drugs, diagnosis, slight lung volume reduction effect, percent predicted postoperative FEV_1_ (%ppoFEV_1_), and DLCO. The findings of this retrospective study should be interpreted with caution due to the limited sample size. While the observed incidence of postoperative complications, such as pneumonia, was low, the small sample size may have precluded a detailed analysis of potential causal relationships and specific underlying mechanisms. Given that inhaled corticosteroids are known to increase the risk of pneumonia, a larger cohort would be beneficial for a more definitive assessment of the safety profile associated with this triple therapy. Future studies with larger sample sizes are needed to confirm these findings and to explore the effects in lung cancer patients with poor pulmonary function. A larger sample size would also allow for subgroup analyses to identify potential effect modifiers. This study prescribed FF/UMEC/VI via the Ellipta inhaler [34]. Many other inhaled drugs and inhalation devices are used in clinical practice, and many studies have reported on each of these interventions. Other drugs or inhalation devices may be more effective. However, the purpose of this study was not to evaluate differences between inhaled drugs or devices, but instead to examine the specific effects of preoperative intervention. While this study demonstrates improvements in FEV_1_, a limitation is that %ppoFEV_1_ was not calculated, which would provide further valuable insights into surgical feasibility and is a consideration for future research. Regarding the absence of DLCO measurements in all patients, it is important to note that our study included patients with untreated COPD whose FEV_1_ increased to >1.5 L after inhaler therapy. Consistent with the British Thoracic Society guideline, DLCO measurement is not mandatory for patients with FEV_1_ > 1.5 L post-bronchodilator. Therefore, we did not perform DLCO measurements in every case.

## 5. Conclusions

Preoperative single-inhaler triple therapy for lung cancer patients with untreated COPD/ACO and FEV_1_ < 1.5 L suggested better respiratory function and expanded radical resection. Moreover, it was suggested to contribute to maintaining respiratory function equivalent to preoperative levels without increasing the risk of perioperative complications. Therefore, preoperative single-inhaler triple therapy is considered safe and effective.

## Figures and Tables

**Figure 1 cancers-17-01803-f001:**
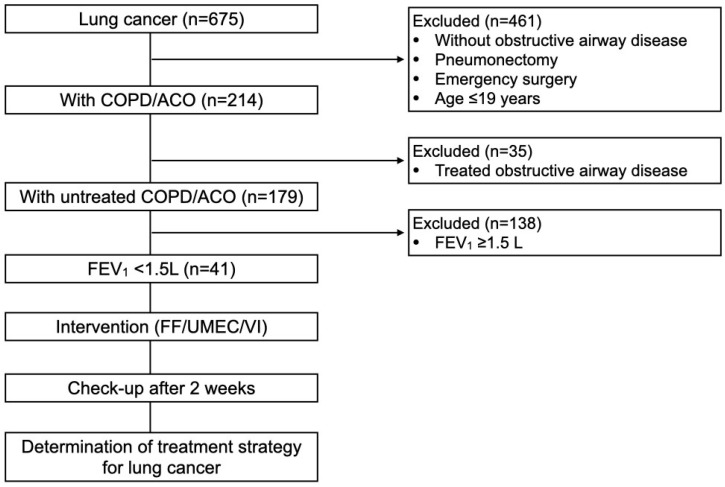
Patient selection flowchart. ACO, asthma and COPD overlap; COPD, chronic obstructive pulmonary disease; FEV1, forced expiratory volume in 1 s; FF/UMEC/VI, fluticasone furoate/umeclidinium/vilanterol.

**Figure 2 cancers-17-01803-f002:**
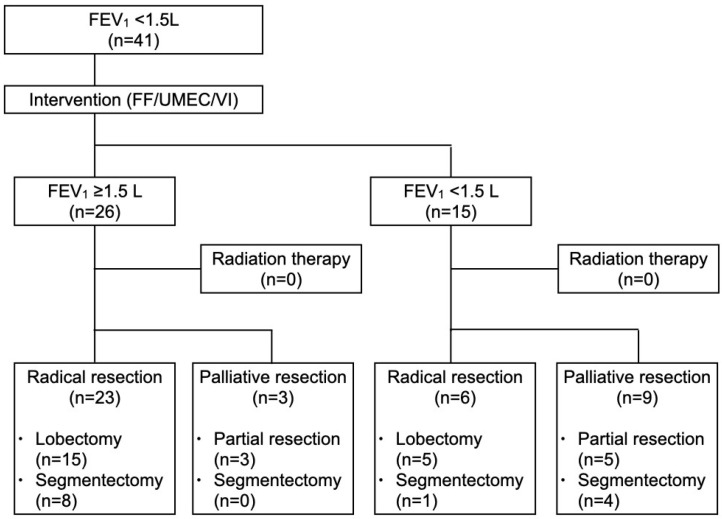
Lung cancer treatment strategy. FEV1, forced expiratory volume in 1 s; FF/UMEC/VI, fluticasone furoate/umeclidinium/vilanterol.

**Figure 3 cancers-17-01803-f003:**
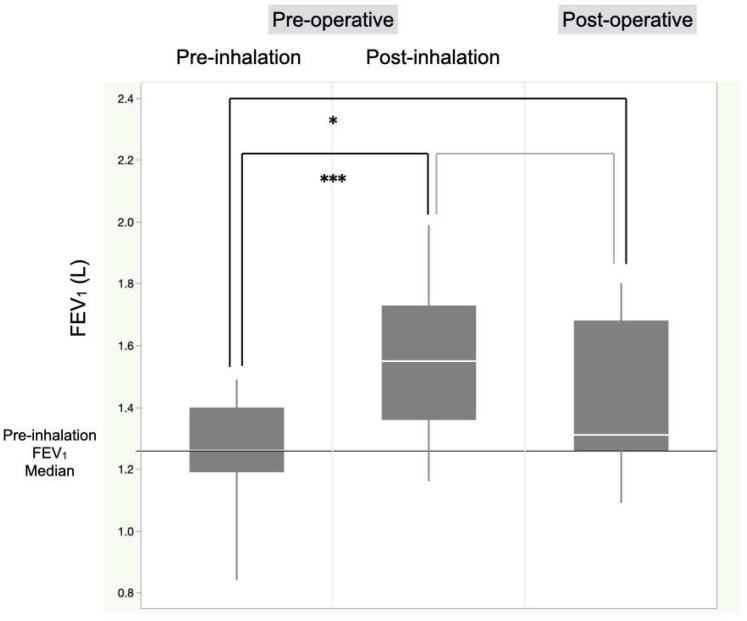
Comparison of perioperative FEV1 levels before and after single-inhaler triple therapy in lung cancer patients who underwent anatomical lung resection. In all patients who underwent anatomical lung resection, there was a significant increase in FEV_1_ post-inhalation therapy compared to that pre-inhalation therapy. The FEV_1_ at 3 months postoperatively was significantly better than pre-inhalation therapy levels. * *p* < 0.005 and *** *p* < 0.001, FEV_1_, forced expiratory volume in 1 s.

**Figure 4 cancers-17-01803-f004:**
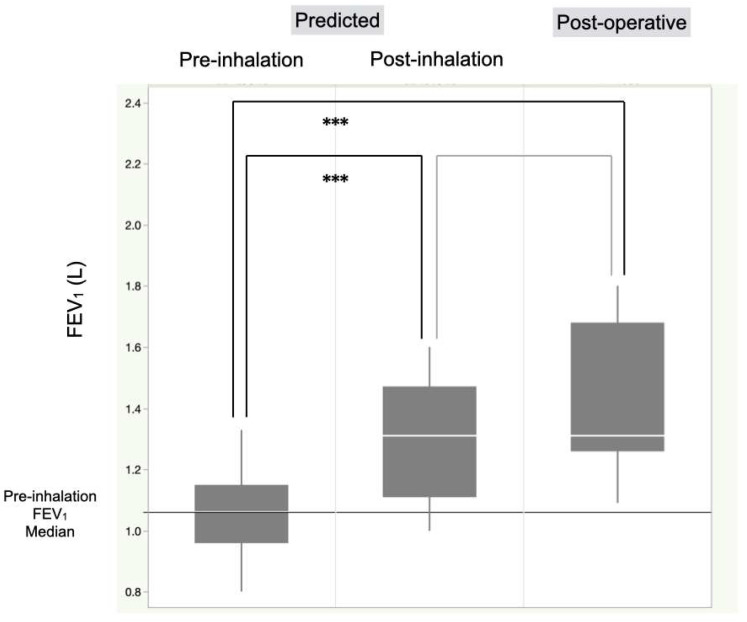
Comparison of predicted FEV1 levels before and after single-inhaler triple therapy in lung cancer patients who underwent anatomical lung resection. The predicted postoperative FEV_1_ pre-intervention was significantly lower than that post-intervention (1.06 vs. 1.31 L, *p* < 0.0001) and the actual FEV_1_ at 3 months postoperatively (1.06 vs. 1.31 L, *p* < 0.0001). *** *p* < 0.001, FEV_1_, forced expiratory volume in 1 s.

**Table 1 cancers-17-01803-t001:** Univariate analysis of lung cancer patients with untreated COPD and ACO.

Characteristic	Post-Interventional FEV1 ≥ 1.5 L (*n* = 26)	Post-Interventional FEV1 < 1.5 L (*n* = 15)	*p*-Value
Age, median [IQR]	77 [73–81]	76 [71–81]	0.49
Sex, male (%)	12 (46.1)	7 (46.7)	0.64
BMI, median [IQR]	22.4 [19.6–25.4]	24.8 [21.5–29.1]	0.091
Hypertension (%)	15 (57.7)	10 (66.7)	0.74
Hyperlipidemia (%)	10 (38.5)	5 (33.3)	0.51
Hyperuricemia (%)	6 (23.1)	3 (20.0)	0.57
Diabetes (%)	11 (42.3)	8 (53.3)	0.53
Smoking history (%)	24 (92.3)	14 (93.3)	0.76
Brinkman index, median [IQR]	840 [435–1010]	1120 [1040–1500]	0.0058
Comorbidities			
Osteoporosis	2 (7.7)	1 (6.7)	0.70
Gastroesophageal reflux	4 (15.4)	2 (13.3)	0.62
Cardiovascular diseases	7 (26.9)	6 (40.0)	0.49
Anxiety and depression	6 (23.1)	3 (20.0)	0.57
Interstitial pneumonia (%)	5 (19.2)	0 (0)	0.14
GOLD classification			
2 (%)	20 (76.9)	10 (66.7)	0.49
3 (%)	6 (23.1)	5 (33.3)	
Spirometry			
FVC (L), median [IQR]	2.29 [2.11–2.42]	2.00 [1.88–2.37]	0.13
%FVC (%), median [IQR]	94.0 [80.8–105.7]	86.7 [74.2–96.3]	0.21
Preint FEV_1_/FVC (%), median [IQR]	61.6 [52.5–66.9]	62.9 [44.3–67.5]	0.77
Postint FEV_1_/FVC (%), median [IQR]	64.9 [52.9–69.2]	65.3 [45.9–69.8]	0.54
Preint FEV_1_ (L), median [IQR]	1.37 [1.26–1.45]	1.14 [1.05–1.32]	0.0050
Postint FEV_1_ (L), median [IQR]	1.63 [1.54–1.76]	1.34 [1.23–1.39]	<0.0001
Postint—Preint FEV_1_(L), median [IQR]	0.29 [0.13–0.52]	0.14 [0.03–0.28]	0.028
Diseased side, right (%)	13 (50.0)	5 (33.3)	0.35
Tumor size (mm), median [IQR]	25 [15–32]	18 [14–23]	0.24
Clinical N1/2 (%)	2 (7.7)	2 (13.3)	0.61
Location, upper segments (%)	7 (26.9)	7 (46.7)	0.31
Radical surgery (%)	23 (88.5)	6 (40.0)	0.0030
Surgical approach			
Uniportal VATS (%)	9 (34.6)	6 (40.0)	0.75
Multiportal VATS (%)	15 (57.7)	9 (60.0)	0.68
Thoracotomy (%)	2 (7.7)	0 (0)	0.52
Procedure			
Partial resection (%)	3 (11.5)	5 (33.3)	0.12
Segmentectomy (%)	8 (30.8)	5 (33.3)	0.70
Lobectomy (%)	15 (57.7)	5 (33.3)	0.19
Number of resected segments (*n*), median [IQR]	3.0 [2.0–4.0]	2.5 [1.8–4.0]	0.64
Intraoperative bleeding (mL), median [IQR]	20 [1–50]	30 [1–50]	0.86
Operative time (min), median [IQR]	157 [131–226]	158 [82–170]	0.21
Chest tube duration (days), median [IQR]	1 [1–1]	1 [1–2]	0.71
Complications (%)			
Total	3 (11.5)	5 (33.3)	0.12
Prolonged air leak	1 (3.9)	1 (6.7)	0.87
Pneumonia	0 (0)	2 (13.3)	0.13
Arrhythmia	1 (3.9)	2 (13.3)	0.54
Atelectasis	0 (0)	0 (0)	N/A
Delirium	0 (0)	0 (0)	N/A
Others	2 (7.7)	2 (13.3)	0.61
Postoperative hospitalization (days), median [IQR]	6 [5–8]	8 [5–10]	0.28
Postoperative FEV_1_, median [IQR]	1.62 [1.31–1.80]	1.25 [1.11–1.26]	0.0014

COPD, chronic obstructive pulmonary disease; ACO, Asthma–COPD overlap; IQR, interquartile range; BMI, body mass index; GOLD, Global Initiative for Chronic Obstructive Lung Disease; FVC, forced vital capacity; FEV1, forced expiratory volume in 1 s; Postint, post interventional; Preint, pre interventional; VATS, video-assisted thoracoscopic surgery; N/A, not applicable.

**Table 2 cancers-17-01803-t002:** Perioperative respiratory function of anatomical lung resection patients with untreated COPD and ACO.

Characteristic	Post-Interventional FEV1 ≥ 1.5 L (*n* = 23)	Post-Interventional FEV1 < 1.5 L (*n* = 10)	*p*-Value
Age, median [IQR]	77 [73–81]	76 [71–81]	0.49
Sex, male (%)	12 (46.1)	7 (46.7)	0.64
BMI, median [IQR]	22.4 [19.6–25.0]	24.4 [22.2–30.5]	0.057
Hypertension (%)	14 (60.8)	8 (80.0)	0.43
Hyperlipidemia (%)	10 (43.5)	3 (30.0)	0.70
Hyperuricemia (%)	6 (26.1)	2 (20.0)	0.54
Diabetes (%)	10 (43.5)	5 (50.0)	0.53
Smoking history (%)	21 (91.3)	9 (90.0)	0.68
Brinkman index, median [IQR]	840 [305–1000]	1100 [990–1550]	0.013
Comorbidities			
Osteoporosis	2 (8.7)	1 (10.0)	0.79
Gastroesophageal reflux	4 (17.4)	2 (20.0)	0.75
Cardiovascular diseases	7 (30.4)	5 (50.0)	0.43
Anxiety and depression	6 (26.1)	1 (10.0)	0.39
Interstitial pneumonia (%)	4 (17.4)	0 (0)	0.29
GOLD classification			
2 (%)	18 (78.3)	7 (70.0)	0.67
3 (%)	5 (21.7)	3 (30.0)	
Spirometry			
FVC (L), median [IQR]	2.32 [2.10–2.50]	1.99 [1.86–2.43]	0.075
%FVC (%), median [IQR]	93.7 [81.6–104.5]	87.4 [73.9–94.8]	0.26
FEV1/FVC (%), median [IQR]	61.5 [53.4–67.3]	59.3 [43.3–67.9]	0.62
Postint FEV1/FVC (%), median [IQR]	64.8 [52.2–70.3]	66.5 [45.8–72.2]	0.93
FEV1 (L), median [IQR]	1.37 [1.26–1.44]	1.13 [1.04–1.27]	0.0013
Postint FEV1 (L), median [IQR]	1.55 [1.47–1.74]	1.33 [1.24–1.37]	<0.0001
Postint—Preint FEV1(L), median [IQR]	0.30 [0.14–0.53]	0.18 [0.08–0.31]	0.11
ppo-Preint FEV1 (L), median [IQR]	1.08 [0.95–1.16]	0.96 [0.83–1.05]	0.016
ppo-Postint FEV1 (L), median [IQR]	1.22 [1.14–1.35]	1.03 [0.98–1.11]	< 0.0001
Diseased side, right (%)	11 (47.8)	3 (30.0)	0.46
Tumor size (mm), median [IQR]	25 [16–32]	21 [15–31]	0.57
Clinical N1/2 (%)	2 (8.7)	2 (20.0)	0.61
Location, upper segments (%)	6 (26.1)	4 (40.0)	0.44
Radical surgery (%)	23 (100.0)	6 (60.0)	0.0051
Surgical approach			
Uniportal VATS (%)	8 (34.8)	3 (30.0)	0.56
Multiportal VATS (%)	14 (60.9)	7 (70.0)	0.71
Thoracotomy (%)	1 (4.4)	0 (0)	0.69
Procedure			
Segmentectomy (%)	8 (34.8)	5 (50.0)	0.46
Lobectomy (%)	15 (65.2)	5 (50.0)	
Number of resected segments (n), median [IQR]	3.0 [2.0–4.0]	2.5 [1.8–4.0]	0.64
Intraoperative bleeding (mL), median [IQR]	20 [1–50]	50 [28–64]	0.089
Operative time (min), median [IQR]	158 [142–225]	167 [157–189]	0.69
Chest tube duration (days), median [IQR]	1 [1–1]	1 [1–3]	0.11
Complications (%)			
Total	3 (13.0)	5 (50.0)	0.036
Prolonged air leak	1 (4.4)	1 (10.0)	0.52
Pneumonia	0 (0)	2 (20.0)	0.085
Arrhythmia	1 (4.4)	2 (20.0)	0.21
Atelectasis	0 (0)	0 (0)	N/A
Delirium	0 (0)	0 (0)	N/A
Others	2 (8.7)	2 (20.0)	0.57
Postoperative hospitalization (days), median [IQR]	6 [5–8]	10 [5–11]	0.13
Postoperative FEV1, median [IQR]	1.61 [1.30–1.78]	1.26 [1.12–1.30]	0.0040

COPD, chronic obstructive pulmonary disease; ACO, Asthma–COPD overlap; IQR, interquartile range; BMI, body mass index; GOLD, Global Initiative for Chronic Obstructive Lung Disease; FVC, forced vital capacity; FEV_1_, forced expiratory volume in 1 s; Postint, post interventional; Preint, pre interventional; ppo, predicted postoperative; VATS, video-assisted thoracoscopic surgery; N/A, not applicable.

## Data Availability

The datasets used and/or analysed in the current study are available from the corresponding author upon reasonable request after approval from the Ethical Committee of the Joetsu General Hospital. The study period ends 3 months to 3 years after the publication.

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
