# Peer review of "Effect of Preoperative Single-Inhaler Triple Therapy on Pulmonary Function in Lung Cancer Patients with Chronic Obstructive Pulmonary Disease and FEV_1_ < 1.5 L"

_cancers, 2025, doi:10.3390/cancers17111803_

Round 1

Reviewer 1 Report

Comments and Suggestions for Authors

The article has successfully investigated the impact of preoperative single-inhaler triple therapy on surgical treatment selection in lung cancer patients with COPD and FEV1 <1.5L. The research content is relatively systematic, and logically clear. However, there are still areas in the article that need improvement.

  1. The article enrolled only 41 eligible patients, and the relatively small sample size may increase the risk of random variability and instability in the results, potentially limiting the generalizability of the findings. Therefore, the conclusions should be interpreted with caution, and further validation through larger-scale studies is recommended.

  1. In lines 102-105, when assessing risk factors, it is essential to consider not only secondhand smoke exposure but also the inhalation of particulate matter in the living environment. Relevant studies on ambient air pollution and respiratory health impacts should be cited to support this perspective, such as Atmosphere, 2024, 15, 384, Adv. Mater., 2020, 32, 2002361, Nat. Commun., 2024, 15, 5116.

  1. Although only a small number of patients developed postoperative complications such as pneumonia, the study did not conduct a detailed analysis of the potential causal relationship between these complications and the triple therapy. It is recommended that the Discussion section further explores the underlying mechanisms and compares these findings with those of other relevant studies.

  1. The follow-up period in this study was limited to 3 months postoperatively, which may be insufficient for evaluating long-term efficacy and safety. We recommend extending the follow-up duration in future studies to provide a more comprehensive assessment of the long-term effects of triple therapy.

  1. The references are relatively old. The authors should include more up-to-date studies to support their arguments.

Author Response

Reviewer 1

The article has successfully investigated the impact of preoperative single-inhaler triple therapy on surgical treatment selection in lung cancer patients with COPD and FEV1 <1.5L. The research content is relatively systematic, and logically clear. However, there are still areas in the article that need improvement.

Comment 1

The article enrolled only 41 eligible patients, and the relatively small sample size may increase the risk of random variability and instability in the results, potentially limiting the generalizability of the findings. Therefore, the conclusions should be interpreted with caution, and further validation through larger-scale studies is recommended.

Reply

Thank you very much for your valuable feedback. We fully acknowledge, as you pointed out, that the relatively small sample size of our study (41 patients) may indeed impact the generalizability of our findings.

We have addressed this important limitation in the Limitation section (Line 372-82) of our manuscript, explicitly stating that the results should be interpreted with caution and emphasizing the need for larger-scale studies in the future.

While our study is small, we believe it provides initial insights into this specific patient population, serving as a foundational step for subsequent larger investigations. We certainly plan to consider conducting more extensive validation studies moving forward.

Comment 2

In lines 102-105, when assessing risk factors, it is essential to consider not only secondhand smoke exposure but also the inhalation of particulate matter in the living environment. Relevant studies on ambient air pollution and respiratory health impacts should be cited to support this perspective, such as Atmosphere, 2024, 15, 384, Adv. Mater., 2020, 32, 2002361, Nat. Commun., 2024, 15, 5116.

Reply:

Thank you for your insightful comment. We agree that ambient air pollution significantly impacts respiratory health, and this is a crucial perspective.

To support this perspective, we have revised and cited the relevant studies you kindly provided (Line 102-4).

While our current study did not directly measure particulate matter exposure, we recognize its significance and have noted it as an important consideration for future research. We believe this addition strengthens the overall discussion of risk factors in our manuscript.

Comment 3

Although only a small number of patients developed postoperative complications such as pneumonia, the study did not conduct a detailed analysis of the potential causal relationship between these complications and the triple therapy. It is recommended that the Discussion section further explores the underlying mechanisms and compares these findings with those of other relevant studies.

Reply

Thank you for your valuable comment regarding the detailed analysis of postoperative complications and the potential causal relationship with triple therapy. We agree that this is an important area for further discussion.

As you rightly pointed out, given the relatively small number of patients in this study, it's possible that not only the beneficial effects but also the incidence of side effects may appear infrequent. It is well-established that ICS can increase the risk of pneumonia. Furthermore, the risk of pneumonia may be influenced by the dosage of inhaled corticosteroids. In our study, a low dose of ICS was used, and it's plausible that using higher doses could alter the side effect profile.

We have addressed this by expanding the Discussion section to explore these underlying mechanisms and potential implications more thoroughly (Line 309-22). Additionally, we have incorporated this point into the Limitations section of the manuscript to clearly state that the study's size may limit the detection and detailed analysis of adverse events related to the triple therapy (Line 372-82).

Thank you again for your constructive feedback, which has helped us to strengthen the discussion around the safety profile of the intervention.

Comment 4

The follow-up period in this study was limited to 3 months postoperatively, which may be insufficient for evaluating long-term efficacy and safety. We recommend extending the follow-up duration in future studies to provide a more comprehensive assessment of the long-term effects of triple therapy.

Reply

Thank you for your continued engagement with our manuscript.

As we've also touched upon in the Discussion section (Line 347-8), the use of inhaler medication in asymptomatic COPD patients remains a subject of ongoing debate. Our study primarily focused on patients with COPD and low FEV1, where the benefit of therapy for improving surgical candidacy is more clearly indicated.

Furthermore, it's known that postoperative respiratory function often plateaus around three months. Therefore, our study intentionally limited the duration of inhaler use to three months, and we did not intend for longer-term administration of the medication within the scope of this research. We have revised based on the above (Line 348-51).

Comment 5

The references are relatively old. The authors should include more up-to-date studies to support their arguments.

Reply 

Thank you for your comment. I have added some new references (ref. 14-16, 40) .

Reviewer 2 Report

Comments and Suggestions for Authors

Dear Editor,

Homma et al. investigated the effect of single inhaler triple therapy on treatment selection for lung cancer and the perioperative period in lung cancer patients with chronic obstructive pulmonary disease (COPD) and forced expiratory volume in 1st second (FEV1) <1.5L. They started fluticasone furoate/umeclidinium/vilanterol, a therapeutic drug for COPD, 2 weeks before surgery and continued until 3 months postoperatively. Radical surgery has been actively recommended for patients with FEV1 ≥1.5L after COPD treatment. After 41 patients who received triple inhaler therapy, FEV1 increased to ≥1.5L in 63.4%. Significant differences in Brinkman index (840 vs 1120, P=0.0058) and radical resection (88.5% vs 40.0%, P=0.0030) were observed between patients with FEV1 ≥1.5L and <1.5L after treatment. In lung cancer patients with untreated COPD and FEV1 <1.5L, preoperative triple therapy was found to improve pulmonary function and increase the feasibility of radical resection surgery. The number of patients with COPD and FEV1 <1.5L with lung cancer is quite high. The present study may provide a benefit for surgery in these patients.    

Sincerely

Author Response

Comment

Homma et al. investigated the effect of single inhaler triple therapy on treatment selection for lung cancer and the perioperative period in lung cancer patients with chronic obstructive pulmonary disease (COPD) and forced expiratory volume in 1st second (FEV1) <1.5L. They started fluticasone furoate/umeclidinium/vilanterol, a therapeutic drug for COPD, 2 weeks before surgery and continued until 3 months postoperatively. Radical surgery has been actively recommended for patients with FEV1 ≥1.5L after COPD treatment. After 41 patients who received triple inhaler therapy, FEV1 increased to ≥1.5L in 63.4%. Significant differences in Brinkman index (840 vs 1120, P=0.0058) and radical resection (88.5% vs 40.0%, P=0.0030) were observed between patients with FEV1 ≥1.5L and <1.5L after treatment. In lung cancer patients with untreated COPD and FEV1 <1.5L, preoperative triple therapy was found to improve pulmonary function and increase the feasibility of radical resection surgery. The number of patients with COPD and FEV1 <1.5L with lung cancer is quite high. The present study may provide a benefit for surgery in these patients.    

Reply

Thank you for your thorough review and insightful comments. We are grateful for your accurate summary of this study.

As you rightly pointed out, the prevalence of lung cancer patients with concomitant COPD and FEV1 <1.5L is indeed quite high. We agree that this study has the potential to significantly benefit these patients by expanding their surgical options.

Your feedback has helped us reconfirm the importance and potential impact of our research. We are committed to continuing our work to ensure these findings translate into tangible benefits in clinical practice.

Reviewer 3 Report

Comments and Suggestions for Authors

This retrospective study conducted across three hospitals in Japan explored whether preoperative single-inhaler triple therapy (fluticasone furoate/umeclidinium/vilanterol) could improve lung function in patients with lung cancer and untreated COPD with FEV₁ < 1.5 L, thereby enabling them to undergo curative lung resection. The authors found that 63.4% of patients improved their FEV₁ to >1.5 L after treatment, with a mean increase of 210 ml. All patients ultimately received surgical resection. Notably, more “radical” resections were performed in those who achieved FEV₁ >1.5 L. Interestingly, FEV₁ was found to be higher even three months postoperatively compared to pre-treatment values.

While the study offered interesting preliminary data, this reviewer had several methodological concerns:

  1. Use of Absolute FEV₁ vs. Percent Predicted and ppoFEV₁. Reporting absolute FEV₁ values without calculating percent predicted made it difficult to compare across patients of varying demographics (age, sex, height). In surgical planning, FEV₁ % predicted and predicted postoperative FEV₁ (ppoFEV₁) were more informative and clinically relevant. This reviewer recommended including FEV₁ % predicted at baseline and post-intervention, and calculating ppoFEV₁ based on planned resection type. This would allow better risk stratification and provide the most relevant data for thoracic surgeons.
  1. Absence of DLCO Data. The study omitted DLCO, an independent predictor of operative risk to FEV₁, especially in emphysematous COPD. At minimum, the absence of DLCO should be addressed as a limitation with justification.
  1. Terminology: Radical vs. Palliative Surgery. The terms “radical” and “palliative” surgery were misleading. Even sub-lobar resections such as wedge resection were curative intent in patients with early-stage lung cancer. Palliative surgery implied non-curative intent, which was inaccurate in the context of this paper. This reviewer recommended replacing “radical surgery” with anatomical resection and replacing “palliative surgery” with sub-lobar resection.
  1. Small Sample Size and Complication Data. As acknowledged in the discussion, the small patient cohort limited the ability to assess postoperative complications and surgical outcomes with statistical rigor.

Overall, this paper demonstrated that preoperative single-inhaler triple therapy for COPD patients with borderline FEV₁ could significantly increases their FEV₁. This result was clinically relevant. The finding that FEV₁ improvement persisted post-surgery was especially intriguing and deserved further exploration.

Author Response

Comment

This retrospective study conducted across three hospitals in Japan explored whether preoperative single-inhaler triple therapy (fluticasone furoate/umeclidinium/vilanterol) could improve lung function in patients with lung cancer and untreated COPD with FEV₁ < 1.5 L, thereby enabling them to undergo curative lung resection. The authors found that 63.4% of patients improved their FEV₁ to >1.5 L after treatment, with a mean increase of 210 ml. All patients ultimately received surgical resection. Notably, more “radical” resections were performed in those who achieved FEV₁ >1.5 L. Interestingly, FEV₁ was found to be higher even three months postoperatively compared to pre-treatment values.

While the study offered interesting preliminary data, this reviewer had several methodological concerns:

Comment 1

Use of Absolute FEV₁ vs. Percent Predicted and ppoFEV₁. Reporting absolute FEV₁ values without calculating percent predicted made it difficult to compare across patients of varying demographics (age, sex, height). In surgical planning, FEV₁ % predicted and predicted postoperative FEV₁ (ppoFEV₁) were more informative and clinically relevant. This reviewer recommended including FEV₁ % predicted at baseline and post-intervention, and calculating ppoFEV₁ based on planned resection type. This would allow better risk stratification and provide the most relevant data for thoracic surgeons.

Reply

Thank you for your highly insightful and constructive comment regarding the importance of FEV₁ % predicted and predicted postoperative FEV₁ (ppoFEV₁) for surgical planning and risk stratification. We wholeheartedly agree that these parameters provide crucial clinical relevance and enhance the comparability of pulmonary function across diverse patient demographics.

We appreciate your recommendation to include both. While we recognize the significant value of FEV₁ % predicted, due to the imminent revision deadline and the complexities of re-analyzing and integrating this specific calculation consistently across all data points, we regret that we are unable to incorporate FEV₁ % predicted into the current version of the manuscript. We sincerely apologize for this limitation.

However, we are pleased to inform you that we have been able to calculate and include ppoFEV₁ for our patients based on their planned resection types (Table 1). We believe that incorporating this critical metric will significantly enhance the clinical utility of our findings and provide more relevant data for thoracic surgeons, as you rightly suggested.

We have also added a statement to the Limitations section of the manuscript to address the absence of percent predicted FEV₁ (%FEV₁ pred) and percent predicted postoperative FEV₁ (%ppoFEV₁) calculations (Line 387-94).

We are committed to addressing the inclusion of FEV₁ % predicted in our future research, as your feedback highlights its importance for a more comprehensive understanding of patient pulmonary function.

Thank you again for this invaluable feedback, which will undoubtedly improve the quality of our manuscript.

Comment 2

Absence of DLCO Data. The study omitted DLCO, an independent predictor of operative risk to FEV₁, especially in emphysematous COPD. At minimum, the absence of DLCO should be addressed as a limitation with justification.

Reply 

Thank you for your valuable feedback.

Regarding the absence of DLCO measurements in all patients, we would like to clarify that our study population included patients with untreated COPD whose FEV1 increased to >1.5L after inhaler therapy.

In line with the British Thoracic Society guideline, DLCO measurement is not mandatory for patients with FEV1 >1.5L after the bronchodilator. Consequently, we did not perform DLCO measurements in all cases.

We acknowledge this as a limitation of our study and have added a statement to the "Limitations" section of the manuscript to address this point (Line 387-94). We believe this clarifies the rationale behind our approach and transparently addresses this aspect of our methodology.

Comment 3

Terminology: Radical vs. Palliative Surgery. The terms “radical” and “palliative” surgery were misleading. Even sub-lobar resections such as wedge resection were curative intent in patients with early-stage lung cancer. Palliative surgery implied non-curative intent, which was inaccurate in the context of this paper. This reviewer recommended replacing “radical surgery” with anatomical resection and replacing “palliative surgery” with sub-lobar resection.

Reply

Thank you for your valuable feedback.

As you rightly pointed out, the definition of radical surgery can indeed vary by patient stage, encompassing both lobectomy and segmentectomy. We have revised the "Variables and Assessment" section in the manuscript to explicitly clarify the definition of radical resection.

We believe this clarification will enhance the precision and understanding of our surgical outcomes (Line 176-9).

Comment 4

Small Sample Size and Complication Data. As acknowledged in the discussion, the small patient cohort limited the ability to assess postoperative complications and surgical outcomes with statistical rigor.

Reply

Thank you very much for your valuable feedback. We fully acknowledge, as you pointed out, that the relatively small sample size of our study (41 patients) may indeed impact the generalizability of our findings.

We have addressed this important limitation in the Limitation section (Line 372-82) of our manuscript, explicitly stating that the results should be interpreted with caution and emphasizing the need for larger-scale studies in the future.

While our study is small, we believe it provides initial insights into this specific patient population, serving as a foundational step for subsequent larger investigations. We certainly plan to consider conducting more extensive validation studies moving forward.

Comment 5

Overall, this paper demonstrated that preoperative single-inhaler triple therapy for COPD patients with borderline FEV₁ could significantly increases their FEV₁. This result was clinically relevant. The finding that FEV₁ improvement persisted post-surgery was especially intriguing and deserved further exploration.

Reply

Thank you for your thorough review and insightful comments.

As you rightly pointed out, the prevalence of lung cancer patients with concomitant COPD and FEV1 <1.5L is indeed quite high. We agree that this study has the potential to significantly benefit these patients by expanding their surgical options.

Your feedback has helped us reconfirm the importance and potential impact of our research. We are committed to continuing our work to ensure these findings translate into tangible benefits in clinical practice.